# A Novel Heterogeneous Parallel System Architecture Based EtherCAT Hard Real-Time Master in High Performance Control System

Hongzhe Shi, Weiyang Lin *, Chenlu Liu and Jinyong Yu

Research Institute of Intelligent Control and Systems, Harbin Institute of Technology, Harbin 150001, China
* Correspondence: wylin@hit.edu.cn

**Abstract:** EtherCAT is one of the preferred real-time Ethernet technologies. However, EtherCAT is not applicable in high-end control fields due to real-time constraints. Clock synchronization and cycle time are the most representative limitations. In this paper, a novel *Heterogeneous Parallel System Architecture* (*HPSA*) with features of parallel computation and hard real-time is presented. An HPSA-based EtherCAT hard real-time master is developed to significantly improve clock synchronization and shorten cycle time. Traditional EtherCAT masters feature serial processing and run on a PC. This HPSA-based master consists of two parts: EtherCAT master stack (EMS) and EtherCAT operating system (EOS). EMS implements the parallel operation of EtherCAT to realize the shorter cycle time, and EOS brings a hard real-time environment to the HPSA-based master to improve clock synchronization. Furthermore, this HPSA-based master operates on a heterogeneous System-on-a-chip (SoC). EMS and EOS form a heterogeneous architecture inside this SoC to achieve low-latency process scheduling. Experimental results show that in our HPSA-based EtherCAT hard real-time master, the cycle time reaches the sub-50 μs range, and the synchronization error reduces to several nanoseconds. Thus, this HPSA-based master has great application value in high-performance control systems.

**Keywords:** EtherCAT master; heterogeneous parallel system architecture; clock synchronization; cycle time



## 1. Introduction

Industrial Ethernet technology is an extension of Ethernet technology. This technology is gradually replacing the traditional Fieldbus technology due to its compatibility, low cost, flexible topology, and high bandwidth. Among the types of Ethernet, the industrial real-time Ethernet is steadily gaining more and more popularity. There are five main industrial real-time Ethernet networks: Powerlink, PROFINET, SERCOS III, Ethernet/IP, and EtherCAT [1]. Table 1 displays their performance parameters, and we can see that EtherCAT is one of the best real-time Ethernet networks.

**Table 1.** Comparison of five real-time Ethernet technologies.

| Performance | Real-Time Ethernet Technologies | | | | |
| --- | --- | --- | --- | --- | --- |
| | Powerlink | PROFINET | SERCOS III | Ethernet/IP | EtherCAT |
| Synchronization error | «1 μs | 1 μs | <1 μs | <1 μs | «1 μs |
| Cycle time | 100 μs | 1 ms | 25 μs | 100 μs | 100 μs |
| Communication scope | 100 m | 100 m | 40 m | 100 m | 100 m |
| Transmission speed | 100 M | 100 M | 100 M | 100 M | 100 M |
| Application cost | cheap | expensive | expensive | cheap | cheap |

Only the EtherCAT master with low latency and high synchronization can make control systems realize rapid and accurate operations. With the continuous development of high-end control fields, traditional EtherCAT masters cannot meet the performance requirements of high-end control systems due to real-time constraints. Therefore, a new EtherCAT master with higher real-time performance is required to facilitate its application in advanced industrial applications.

Cycle time and clock synchronization are the most important indexes of EtherCAT real-time performance. Cycle time refers to the frequency of EtherCAT frames sent by the EtherCAT master. The traditional EtherCAT master cannot achieve a short cycle time, which hinders the high-speed operation of the control system. All controlled objects in EtherCAT systems are expected to operate simultaneously, so clock synchronization reflects the operating accuracy of the control system. The traditional EtherCAT master has limited effect in compensating synchronization errors, which prevents real-time optimization.

*Simple Open-Source EtherCAT Master (SOEM)* and *IgH* are existing open-source EtherCAT master stacks. The literature [2] designed a data transceiver algorithm to increase the stability of EtherCAT masters. In [3], an EtherCAT frame detection function was developed to decrease the frame loss rate. In [4], an exponential moving average (EMA) filter was adopted to reduce synchronization errors in EtherCAT systems. The article [5] enhanced the operating efficiency of EtherCAT masters by optimizing the EtherCAT frame size. Except for the stack optimization, the most common approach is ameliorating the real-time environment of operating platforms, such as installing the *Xenomai* patch in the Linux kernel to provide high real-time support [6] or porting EtherCAT masters to high-performance embedded processors to mitigate system jitter [7–9].

EtherCAT provides a solution called a distributed clocks (DC) mechanism to ensure slave–slave synchronization, so the master–slave clock synchronization has great room for improvement [10,11]. In [12], a master–slave synchronization algorithm was developed to compensate for the propagation delay between the EtherCAT master and the reference slave. The literature [13] used precision time control protocol (PTCP) embedded in EtherCAT frames to achieve high-quality master–slave synchronization. Furthermore, integrating the EtherCAT master with an EtherCAT slave to synchronize the master's clock with the reference clock is a good idea to satisfy high real-time requirements [14,15]. Some research institutions focus on developing EtherCAT peripheral equipments, such as shortening the cycle time by introducing the EtherCAT accelerator [16].

Existing real-time optimization research is mainly based on traditional EtherCAT masters. Therefore, the serial operation mode greatly limits the optimization effects of real-time performance. The operating system of the EtherCAT master is usually a multi-process system. Therefore, multi-thread delays and scheduling conflicts are inevitable, which burden the improvement of clock synchronization. Moreover, on the existing operating platform, the inter-module communication latency between the EtherCAT master and the operating system cannot be effectively solved, which will bring additional synchronization errors.

In this study, we propose a novel system architecture called HPSA and develop an HPSA-based EtherCAT hard real-time master for high-performance control systems. As shown in Figure 1, there are two different system modules inside this master: EMS and EOS, which work together to bring parallel computing and hard real-time characteristics to the HPSA-based master. The major contributions of this paper are summarized as follows:

1.  We use hardware description language (HDL) to create a novel EMS. This EMS features data parallelism and conflict-free processing. In this way, the HPSA-based master can operate more efficiently and stably to achieve a lower cycle time and reduce the frame loss rate.
2.  We build an EMS-based EOS. This EOS supplies a stable and hard real-time operating environment for the HPSA-based master to mitigate the process scheduling latency and achieve high-speed operation of complex control algorithms.

3.    We place this HPSA-based master into EtherCAT systems. Experimental results demonstrate that the HPSA-based master can operate well. Compared with traditional EtherCAT masters, the performance of clock synchronization and cycle time have significantly improved.

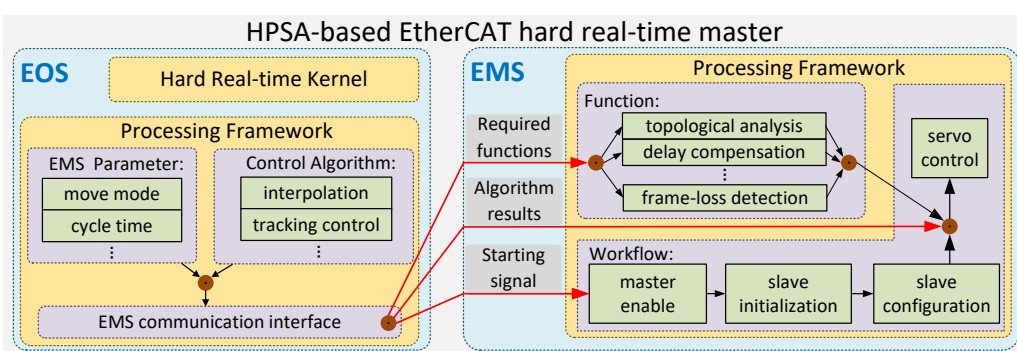

**Figure 1.** Heterogeneous parallel system architecture of EtherCAT hard real-time master.

The remainder of this paper is as follows. Section 2 presents related work. Section 3 introduces the component modules of EMS. Section 4 describes the creating process of EOS. Section 5 presents the framework of the HPSA-based master. Section 6 conducts related experiments and analyzes experimental results. Finally, Section 7 draws final conclusions.

## 2. Related Work

### 2.1. Cycle Time Optimization

Most Ethernet technologies are not suitable for transmitting complex information in a short period [17,18] because the Ethernet system must manage multiple frames in each cycle to ensure that all devices respond to commands. For EtherCAT, there is a unique frame transmission mechanism called "on the fly" to solve this deficiency. As shown in Figure 2, a typical EtherCAT system consists of two parts: EtherCAT master and EtherCAT slave. Computers with Ethernet interfaces or embedded devices with Ethernet controllers can act as EtherCAT masters, and devices with the EtherCAT slave controller (ESC) chip are regarded as EtherCAT slaves [19,20]. According to the "on the fly" mechanism, the EtherCAT master only sends one EtherCAT frame in each cycle, and all EtherCAT slaves share this frame. When this frame arrives at each slave, the ESC extracts information, executes commands, and feedbacks operation results. This frame returns to the EtherCAT master after traversing all slaves to help the master obtain the running status of each slave.

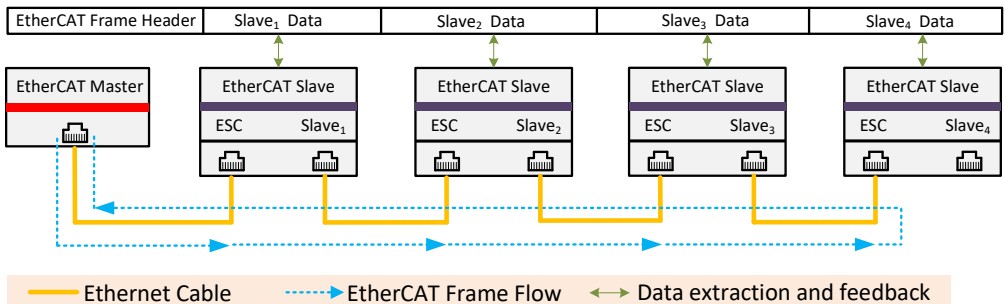

**Figure 2.** EtherCAT system and "on the fly" mechanism.

Thanks to this mechanism, the maximum transmission speed of EtherCAT can even exceed 100 Mbps, and the cycle time is below 100 μs [21,22]. However, in practical control applications, only a few EtherCAT systems can achieve less than 100 μs cycle time [23]. Therefore, optimizing the EtherCAT master to shorten the cycle time is a key point in EtherCAT real-time improvement.

One work of this paper is to optimize the cycle time of the EtherCAT master. A new EMS is created to enable the HPSA-based master to run in parallel, thus improving operational efficiency and shortening cycle time. Meanwhile, this EMS adopts the method of modular design to realize conflict-free invocation of all processes, thus enhancing stability and reducing frame loss rate during operation.

### 2.2. Clock Synchronization Improvement

In addition to the cycle time, another prominent feature of EtherCAT is clock synchronization. The purpose of clock synchronization is to make all EtherCAT slaves operate simultaneously. The clock synchronization error is mainly caused by the EtherCAT frame transmission delay, EtherCAT slave initial offset, and EtherCAT slave crystal jitter. EtherCAT provides the DC mechanism to ensure the synchronization error between EtherCAT slaves is less than 1 μs [24–26].

In addition to the EtherCAT master, the real-time performance of the operating system is also important. The process scheduling latency caused by the system jitter will affect the operation of the EtherCAT master and amplify the synchronization error. Therefore, a stable and real-time working environment plays a critical role in clock synchronization.

Another work is improving the clock synchronization of the EtherCAT system. In addition to the DC mechanism of the EtherCAT master, a stable and real-time working environment plays a critical role in the synchronization accuracy improvement. An EOS is built to supply a hard real-time operating environment to reduce the clock jitter. This EOS is built based on EMS, which can alleviate the process scheduling delay. Furthermore, EMS and EOS inside the heterogeneous SoC form a heterogeneous architecture to realize inter-module low-latency communication.

## 3. Composition of EMS

EMS is the implementation mechanism of EtherCAT masters. Figure 3 shows the structure and workflow of the EMS created in this design. This EMS runs in parallel to improve operating efficiency and shorten cycle time. We adopt the modular design method to divide this EMS into four trunk modules and separate each module into multiple sub-modules according to work content. Each module is independent and can call other modules as needed during operation to avoid resource conflict and achieve low energy consumption.

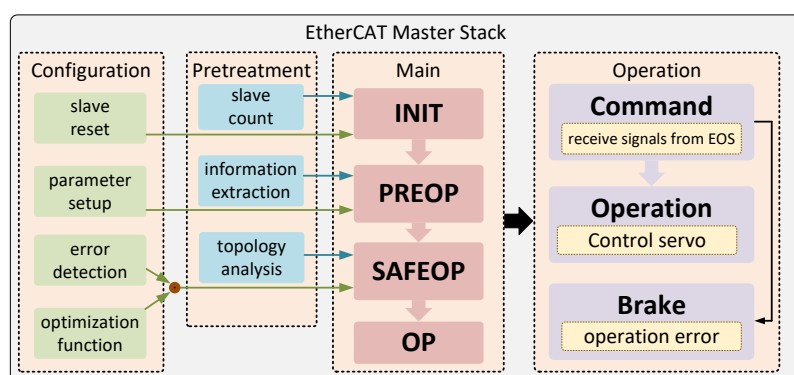

**Figure 3.** Structure and workflow of EtherCAT master stack.

### 3.1. Process Part

The process part describes the workflow of EMS, it includes the main module and the operation module. The main module concentrates on state changes of EtherCAT slaves, and the operation module pays attention to the operation of EtherCAT slaves.

#### 3.1.1. Main Module

EMS first enters the main module after startup. The main module consists of four state sub-modules corresponding to the four configuration states of EtherCAT slaves:

initialization state (INIT), pre-operation state (PREOP), safe-operation state (SAFEOP), and operation state (OP). The EtherCAT master must complete specified configuration in each state sub-module. In INIT, EMS activates the ESC to make each EtherCAT slave in the EtherCAT system work properly. In PREOP, EMS fetches hardware characteristics of EtherCAT slaves such as electrical specification and register address, then configures devices reasonably according to these parameters. In SAFEOP, EMS first checks these EtherCAT slaves. If there are any problems during configuration, devices will return to INIT. Then, EMS executes functions or mechanisms EtherCAT slaves supported for performance optimization. In OP, all preparations are complete, EMS enters the operation module.

Each state sub-module calls other sub-modules to achieve desired operations. The benefit of this model is to simplify EMS and achieve conflict-free resource allocation. When different states require the same function, they only need to call the specified sub-module in turn.

### 3.1.2. Operation Module

In the operation module, EtherCAT slaves run according to instructions from the EtherCAT master. In the running process, EMS receives signals from EOS and encapsulates them into EtherCAT frames. Furthermore, EMS receives feedback information from slaves to judge whether these devices work as desired. We add the emergency brake function to ensure the security of EtherCAT systems. EMS triggers the brake to prevent systems from losing control when the running state of EtherCAT slaves deviate from expectations.

### *3.2. Function Part*

This part consists of the pretreatment module and the configuration module to help the main module realize the state conversion of EtherCAT slaves. The pretreatment module offers initialization and preparation for EtherCAT slaves. The configuration module realizes register management and performance optimization. Function extension of EMS can be realized by modifying this part.

### 3.2.1. Pretreatment Module

In INIT, EMS first counts the number of EtherCAT slaves in systems. This step is necessary. As shown in Figure 2, the length of EtherCAT frames adjusts according to the number of devices. If this step is lost or wrong, slaves can not respond to frames, and all operations later will be invalid. Since the hardware characteristics of each EtherCAT slave are different, in PREOP, EMS should extract device information before configuration to ensure accurate management. The last task is EtherCAT topology analysis. The EtherCAT topology refers to the distribution of each EtherCAT slave in EtherCAT systems, which can help us find the reference slave and calculate the frame transmission delay. Thus, this task concerns the clock synchronization accuracy.

### 3.2.2. Configuration Module

Before configuration, EMS needs to reset all EtherCAT slaves to avoid troubles caused by previous operation results. In PREOP, EMS configures EtherCAT slaves according to the information fetched in the pretreatment module. In SAFEOP, EMS calls optimization functions as required to improve the performance of EtherCAT systems. For example, EMS can calculate the initial offset of slave n $T_{offset}(n)$ through the local clock of slave n $t_{local}(n)$ and the local clock of the reference slave $t_{sys\_ref}$. Through Equations (1), the initial offset of slave n can be compensated, which enhances the clock synchronization between the slave n and the reference slave.

$$T_{offset}(n) = t_{local}(n) - t_{sys\_ref} \tag{1}$$

## 4. Creation of EOS

As shown in Figure 1, the HPSA-based master needs an operating system (OS) to offer a real-time operation environment. This OS must match EMS exactly to avoid system crashes due to application incompatibility. In this design, EOS is created based on EMS to alleviate system jitter and reduce the response delay. Thus, the synchronization accuracy of EtherCAT systems can be greatly improved.

### 4.1. The Hard Real-Time Characteristic Acquisition of EOS

The real-time OS is a kind of OS that emphasizes real-time performance. Real-time OS can divide into two categories according to the severity of timeout results. The hard real-time OS does not allow timeout, and all tasks must finish within the specified time, or the system will crash. The soft real-time OS is not strict with the timeout, and the task's timeout will not affect the system operation. Thus, the hard real-time OS provides a better real-time environment. Linux is a typical soft real-time OS and can be converted to the hard real-time OS by installing a real-time kernel patch *PREEMPT_RT* [27,28]. Hence, we use this approach to bring hard real-time behavior to EOS.

### 4.2. Design Flow of Embedded EOS

The embedded operating system is a type of system software that runs on the embedded system. We use Petalinux tools to create an embedded Linux operating system and make it work on a heterogeneous SoC. Petalinux is a toolset, including Linux libraries, u-boot source code, and yocto recipes, which can customize embedded Linux system components for Xilinx embedded processors.

Figure 4 shows the creation process of EOS. The SDK file contains compilation results of EMS, such as memory size and communication address. The working environment of Petalinux is built based on this file to ensure EOS can closely combine with EMS. In addition to the SDK file, Petalinux needs Linux system files to form the system architecture. In this design, we only use the most basic files to create a simple architecture to avoid multi-thread scheduling latency. The Device-tree file describes the hardware information of the operating platform, such as network ports and sd card interfaces in the SoC board. The fsbl file is the loading program to initialize the memory of EOS and configure EMS in the loading process. Linux-xlnx is the Linux kernel source package provided by Xilinx, which can build Linux kernel for embedded processors. The root file system (Rootfs) in Linux pays attention to the management of documents. After the preparatory work, petalinux commands are called to build the embedded Linux EOS. The created EOS only has the soft real-time feature, so we can transform it into a hard real-time EOS by installing the *PREEMPT_RT* patch.

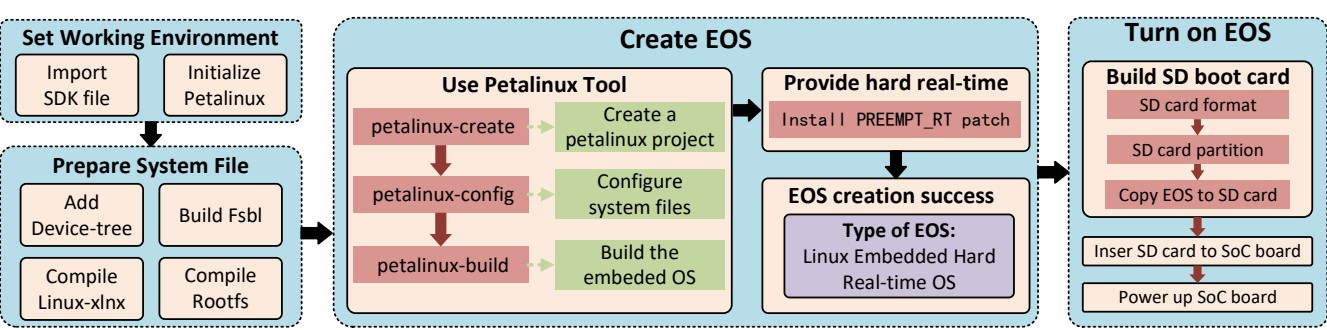

**Figure 4.** Creation and start-up of EtherCAT operating system.

So far, the creation of the Linux-embedded hard real-time operating system is complete, and the next step is to start the EOS on the operation platform. As shown in Figure 4, we chose an sd card as the carrier to load the EOS into the heterogeneous SoC. Figure 5 exhibits the operating platform of the HPSA-based master. EMS and EOS form a heterogeneous architecture inside this SoC to realize the operation of the HPSA-based master.

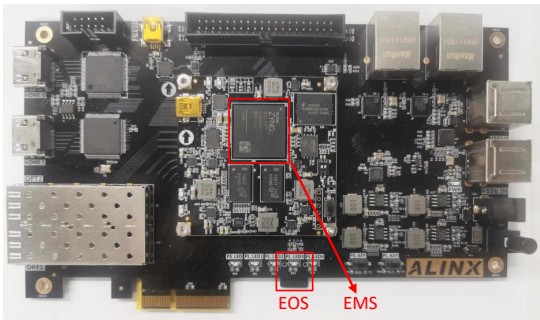

**Figure 5.** Operating platform of HPSA-based master.

## 5. Framework of HPSA-Based Hard Real-Time Master

The HPSA-based master takes a heterogeneous SoC as the operating platform, as shown in Figure 5. This SoC consists of two parts: programmable logic (PL) and processing system (PS). The PL part corresponds to the Field Programmable Gate Array (FPGA), and the PS part is equal to the Advanced RISC Machine (ARM). Thus, this SoC can be regarded as a combination of an FPGA and an ARM. Figure 6 depicts the architecture and composition modules inside this HPSA-based master. All modules in this master exist in the form of IP-core to improve security and portability. EMS runs in the PL part to realize the parallel operation of the EtherCAT master. EOS works in the PS part to provide hard real-time performance and operation commands for the EtherCAT master. Therefore, EMS and EOS form a heterogeneous architecture inside this SoC. Thanks to the heterogeneous nature of the HPSA-based master, EOS transmits signals to EMS through an on-chip bus called AXI, which significantly reduces the process scheduling latency within the system.

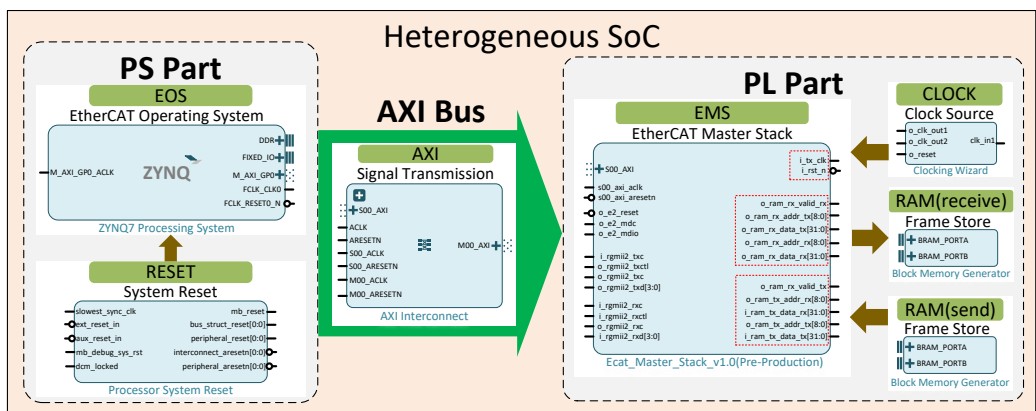

**Figure 6.** Framework of HPSA-based master and communication between modules.

### 5.1. PS Part

The PS part contains EOS and the RESET module. In addition to generating control commands, EOS can implement motion algorithms as needed to apply to different control areas. For example, a kinematic framework can be built in EOS to realize the robotic control of the HPSA-based master. To ensure the security of the HPSA-based master, a RESET module is built to provide the reset function. When the timeout occurs in this hard real-time EtherCAT master, the RESET module can clear the clock disorder and restart EOS in time to avoid terrible consequences.

### 5.2. PL Part

The PL part consists of three modules, CLOCK, RAM, and EMS. The CLOCK module is the clock source of the EMS, which controls the operation period of the HPSA-based master. Furthermore, this module supplies reset signals to prevent EMS from losing control.

There are two kinds of RAM modules that correspond to the two flow directions of EtherCAT frames. The RAM-sending module stores sending frames. In traditional EtherCAT masters, frame loss may occur when the cycle time is too short for EtherCAT frame configuration. In the HPSA-based master, EMS completes EtherCAT frame configuration in advance and stores configured frames in ram. This behavior can reduce the frame loss rate and improve the operational efficiency of the EtherCAT master.

The RAM-receiving module stores feedback frames. In traditional EtherCAT masters, the master may hardly receive feedback frames accurately when running at high speed. Thus, it is difficult for the master to judge the operation status of EtherCAT slaves, which may cause terrible consequences in high-precision applications. In the HPSA-based master, all feedback frames are stored in the ram, and the master can extract the corresponding feedback information as required. This method avoids the loss of feedback information and improves the operation accuracy of the EtherCAT master.

EMS receives control commands and algorithm results from EOS and encapsulates these parameters into EtherCAT frames to control the operation of EtherCAT slaves. EMS should invoke various functional modules to meet the performance requirements of high-end control systems. In addition, EMS must adjust the current operation status in real time according to the feedback information to ensure the high precision of the HPSA-based master.

## 6. Experimental Results

After the HPSA-based hard real-time master is created, we put this master into an EtherCAT system to test whether it can work properly. Then, the impact of the HPSA-based master on the real-time performance of EtherCAT systems is analyzed by measuring the cycle time and clock synchronization.

### 6.1. Experimental Setup

Figure 7 illustrates the EtherCAT system, which consists of the HPSA-based master and two kinds of EtherCAT slaves. The EtherCAT I/O modules run on STM32F767 boards to provide signal interfaces for performance tests. The EtherCAT servo motors are ASDA series servo drives, which supply information for running state analysis.

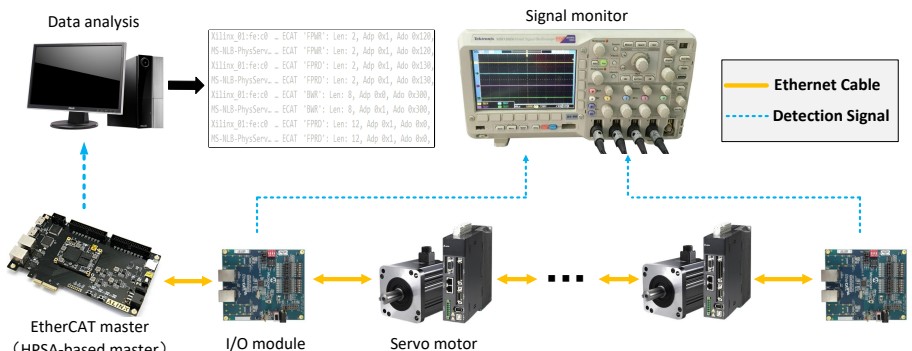

**Figure 7.** Experimental platform of the HPSA-based master.

The Tektronix Oscilloscope connects the sync0 interface provided by EtherCAT I/O modules to test the cycle time and clock synchronization. The sync0 event is a periodic interrupt event that is triggered each time the EtherCAT frame arrives at the EtherCAT slave. Therefore, the trigger frequency of the sync0 event reflects the cycle time of the EtherCAT master. The oscilloscope works in persistence mode to obtain the actual cycle time of the EtherCAT master in real time. Due to the DC mechanism, all EtherCAT slaves expect to trigger sync0 events simultaneously. Therefore, the trigger time difference of sync0 events reflects synchronization errors. The oscilloscope needs to work in persistence mode to master the synchronization status of the EtherCAT system during operation.

The working state of the HPSA-based master can be analyzed through EtherCAT frames. For example, Figure 8 exhibits the EtherCAT frame obtained by wireshark software. It can be seen that the EtherCAT frame can be divided into two categories according to the IP source address, corresponding to sending and receiving. The frame sending interval is 1 ms, which means the HPSA-based master operates in the 1 ms cycle time. Furthermore, the IP source address of EtherCAT sending frames is Xilinx_01:fe:c0, which is consistent with the IP address of the HPSA-based board. The state feedback by EtherCAT slaves is 0800, which implies this device is in the OP state. Analysis results show that the HPSA-based master can run like traditional EtherCAT masters, so the work of HPSA-based EtherCAT hard real-time master creation is successful.

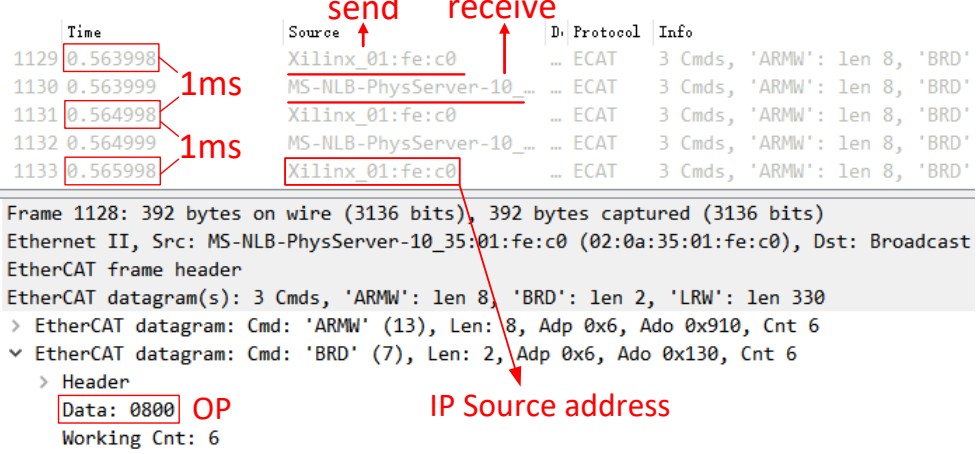

**Figure 8.** Information provided by EtherCAT frame.

### 6.2. Experiment of Cycle Time

Although the cycle time of EtherCAT can reach the sub-100 µs range, most applications are in the 500–1000 µs domain. Thus, the shortest cycle time of EtherCAT masters is a significant indicator in high-performance applications. In this experiment, we measured the cycle time performance of the HPSA-based master and compared the test results with the TwinCAT master. TwinCAT is an abbreviation for the Windows Control and Automation Technology, it is a commercial software to make the PC act as the EtherCAT master.

Figure 9 shows that the shortest interval of the HPSA-based master is about 31.25 µs. Affected by the stability of the EtherCAT master, the actual cycle time will jitter around the set cycle time. The actual cycle time can be calculated through the system time provided by the reference slave in each cycle.

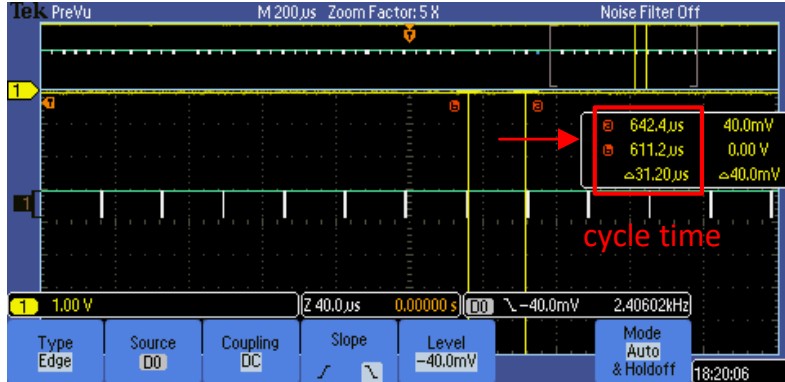

**Figure 9.** Cycle time measurement of the HPSA-based master.

Figure 10a is the actual cycle time of the HPSA-based master during operation. We can see that the operation period is set to 31.25 µs, and the cycle time fluctuates around this

parameter. It is a normal phenomenon because the cycle time cannot be constant due to the system jitter and external interference. The degree of cycle time jitter can be reduced by improving the stability and optimizing the real-time performance of the EtherCAT master. The cycle times of the HPSA-based master are in the 26–35 µs domain, which is acceptable and does not affect the operation of the master.

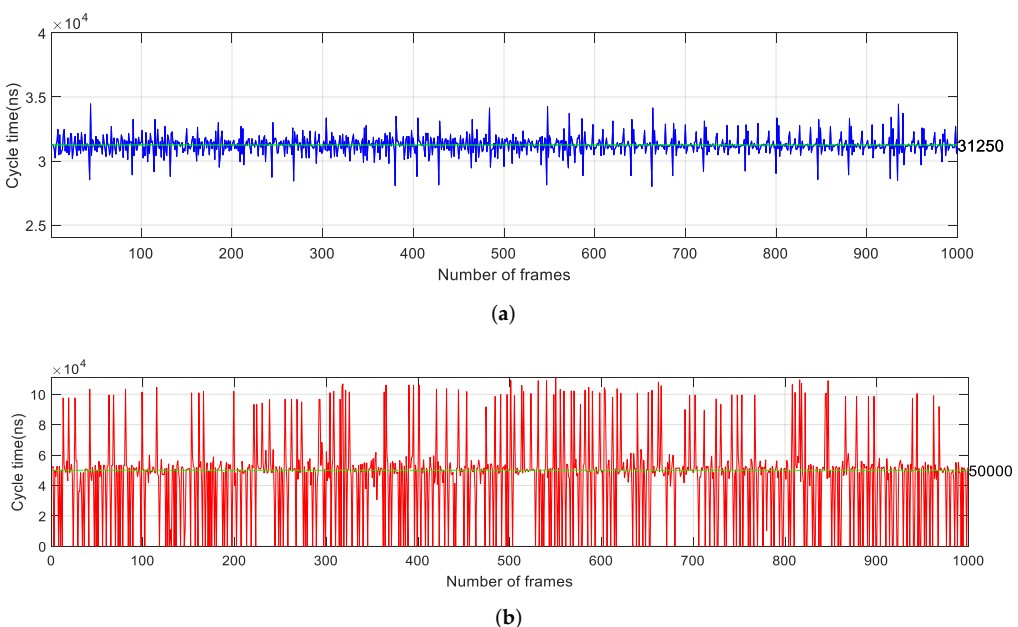

**Figure 10.** Comparison of Cycle time in different EtherCAT masters: (**a**) Cycle time of the HPSA-based master (operation period: 31.25 µs). (**b**) Cycle time of the TwinCAT master (operation period: 50 µs).

Figure 10b is the actual cycle time of the TwinCAT master. Due to performance limitations, the shortest cycle of this master can only reach the 50 µs. It can be seen that compared with the HPSA-based master, the cycle times of the TwinCAT master are in the 0–100 µs domain, which is too large, and the TwinCAT master may hardly operate in this cycle time.

In addition to the jitter range of cycle time, it is necessary to detect whether frame loss occurs during operation. EtherCAT frame loss refers to the phenomenon that the EtherCAT master fails to send frames on time due to performance limitations, which may cause terrible consequences in the high-precision control field. The frame loss rate is related to the real-time performance of the EtherCAT master, so we only need to test it when the master runs in the shortest cycle time. In this experiment, the EtherCAT frame loss rate is measured through the position information fed back by EtherCAT servo motor drivers.

Figure 11a displays the actual position of servo drivers during operation. We collect these position parameters and draw the running trajectory of EtherCAT servo motors. As shown in Figure 11b, the motor moves back and forth at a constant speed. The blue line is the motor running trajectory under the control of the HPSA-based master, which reflects the frame loss rate. It can be seen that this motor operates smoothly and continuously, so there is no EtherCAT frame loss during operation, and the HPSA-based master can operate well in the 31.25 µs cycle time.

The red running trajectory reflects the frame loss rate of the TwinCAT master. Compared with the blue one, there are several interrupts on the track, which means frame loss occurs there. Therefore, the TwinCAT master cannot work normally in the 50 µs cycle time due to insufficient performance.

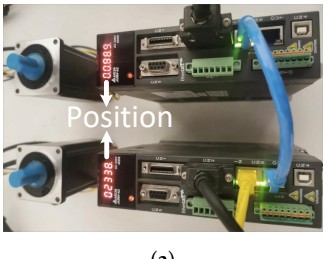
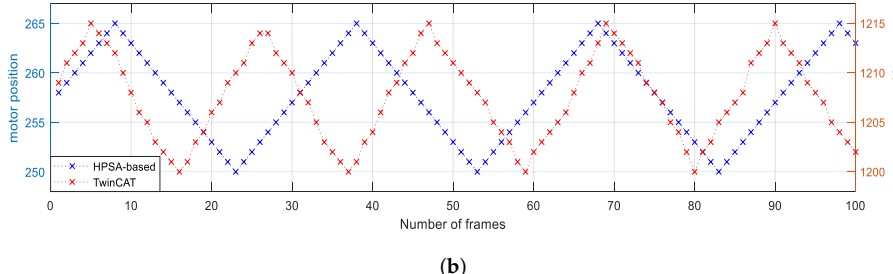

(**a**)                                         (**b**)

**Figure 11.** Position information of EtherCAT servo motors. (**a**) Current position. (**b**) Comparison of running trajectories.

The experimental results of the cycle time are summarized in Table 2. First, it can be seen that the HPSA-based master can reach a shorter cycle time than the TwinCAT master, which means that the master with parallel characteristics can achieve higher work efficiency. Then, it can be obtained from the jitter range of cycle time that the stability of the HPSA-based master is far better than the TwinCAT master, and the frame loss rate of the HPSA-based master is nearly zero. Therefore, the HPSA-based master is more suitable for high-speed control systems.

**Table 2.** Cycle time comparison of EtherCAT masters.

| Cycle Time (µs) | Operation Period (µs) | | | | | |
|---|---|---|---|---|---|---|
| | **1000** | **500** | **250** | **125** | **50** | **31.25** |
| HPSA-based master | 995–1005 | 495–505 | 245–260 | 115–130 | 45–60 | 25–35 |
| TwinCAT master | 990–1010 | 490–515 | 230–270 | 80–160 | 0–100 | - |

*6.3. Experiment of Clock Synchronization*

In this experiment, we test the impact of the HPSA-based master on clock synchronization by measuring the synchronization error between EtherCAT slaves. Figure 12 shows the measurement operation, the top line(0) indicates the sync0 signal of the reference slave, so the maximum time difference between the other slave and the reference slave reflects the synchronization error of this EtherCAT system.

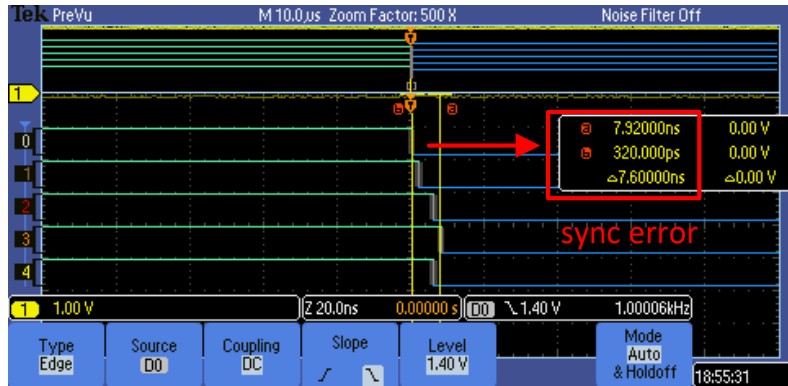

**Figure 12.** Synchronization error measurement of the HPSA-based system.

We take the HPSA-based master to form the HPSA-based system and compare the experimental results with the TwinCAT system. As shown in Figure 13, we carried out two groups of experiments to analyze the real-time performance of the HPSA-based master relative to the TwinCAT master. In each group, we performed six experiments to investigate the influence of the number of slaves and the cycle time on the synchronization accuracy. We sampled 3000 times each experiment to ensure the accuracy of analysis results.

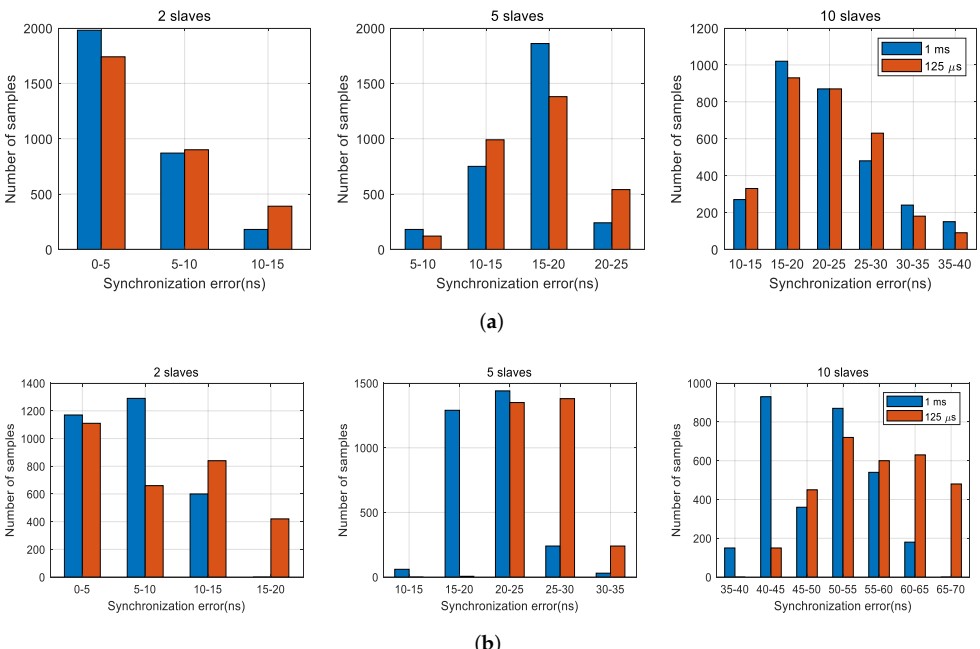

**Figure 13.** Comparison of synchronization error in different EtherCAT systems: (**a**) Synchronization error in HPSA-based system. (**b**) Synchronization error in TwinCAT system.

Figure 13 presents the influence of the number of devices and cycle time on clock synchronization. First, the synchronization error increases with the number of EtherCAT slaves. This phenomenon is inevitable because the compensation accuracy of the DC mechanism is limited, and slight errors will accumulate with the number of devices. Then, the synchronization error increases with the shortening of the cycle time. This problem can alleviate by strengthening the operational efficiency of the EtherCAT master because the DC mechanism can operate more accurately under a high real-time environment.

Figure 13a shows the synchronization error in the HPSA-based system. Most notably, when the master runs in the 1 ms cycle time and 125 μs cycle time, the synchronization accuracy is almost the same. The reason is that the HPSA-based master implements the DC mechanism in parallel to reinforce stability and accuracy, so the cycle time has little effect on clock synchronization.

Figure 13b shows the synchronization error in the TwinCAT system. Compared with the HPSA-based system, it can be seen that the shorter the cycle time, the worse the synchronization accuracy of EtherCAT systems. Furthermore, synchronization errors caused by the number of EtherCAT devices are more serious than the HPSA-based master.

The experimental results of the clock synchronization are summarized in Table 3. Thanks to the high real-time performance of the HPSA-based master, in the HPSA-based system, the impact of the number of EtherCAT slaves on the clock synchronization is greatly reduced over the TwinCAT system. However, compared with the TwinCAT system, the cycle time has little impact on the synchronization accuracy of the HPSA-based system. Thus, this HPSA-based hard real-time master has great application value in high-performance control systems.

**Table 3.** Synchronization error comparison of EtherCAT systems.

| Sync Error (ns) | Cycle Time: 1000 μs | | | Cycle Time: 125 μs | | |
|---|---|---|---|---|---|---|
| | 2 Slaves | 5 Slaves | 10 Slaves | 2 Slaves | 5 Slaves | 10 Slaves |
| HPSA-based system | 0–15 | 5–25 | 10–40 | 0–15 | 5–25 | 10–40 |
| TwinCAT system | 0–15 | 10–35 | 35–65 | 0–20 | 20–35 | 40–70 |

## 7. Conclusions

In this paper, an HPSA-based EtherCAT hard real-time master is presented for high-performance control systems. This HPSA-based master consists of EMS and EOS. EMS brings the parallel processing feature to the HPSA-based master, which enormously shortens the cycle time. EOS supplies a hard real-time environment, which minimizes the synchronization error. EMS and EOS form a heterogeneous architecture inside this HPSA-based master to achieve low latency scheduling. Experimental results show that compared with traditional EtherCAT masters, this HPSA-based EtherCAT hard real-time master can reach a shorter cycle time. Moreover, the clock synchronization of the HPSA-based system is significantly improved.

**Author Contributions:** Conceptualization, H.S. and W.L.; methodology, H.S.; software, H.S. and C.L.; validation, H.S.; formal analysis, H.S. and W.L.; investigation, H.S.; resources, W.L. and J.Y.; data curation, H.S.; writing—original draft preparation, H.S. and C.L.; writing—review and editing, H.S. and W.L.; visualization, H.S.; supervision, W.L. and J.Y.; project administration, W.L. and J.Y.; funding acquisition, W.L. and J.Y. All authors have read and agreed to the published version of the manuscript.

**Funding:** This research was supported in part by the National Natural Science Foundation of China under Grants (62273116) and (61973099).

**Acknowledgments:** The authors are thankful to the anonymous reviewers whose comments helped us improve the paper.

**Conflicts of Interest:** The authors declare no conflict of interest.

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
