# Peer review of "A Novel Heterogeneous Parallel System Architecture Based EtherCAT Hard Real-Time Master in High Performance Control System"

_electronics, doi:10.3390/electronics11193124_

Round 1
Reviewer 1 Report
This paper is a novel Heterogeneous Parallel System Architecture(HPSA) with features of parallel computation and low power consumption is presented, and an HPSA-based EtherCAT hard real-time master is developed to significantly improve clock synchronization and shorten cycle time. The paper is well written and the analyses along with the results are well expressed. These are some of the comments that need to be addressed.
1. The presentation contains several grammar errors; English must be improved too.
2. Abstract: should highlight the main idea of the method; specifically, what is novel.
3. Introduction: needs a stronger motivation.
4. Related work: missing, focus on recent works.
5. Quality of figures and tables are rather poor; some drawings are stretched; some are too small (Fig. 6 , Fig. 8 for example).
6. Experimental results: Comparison results are not clear?
7. Citation is required for equations?
8. Clarify on Framework of HPSA-based Hard Real-time Master?
9. Experimental results: Missing?
10. It looks that this Heterogeneous Parallel System Architecture is Simulated Annealing in disguise. Please explain the differences.
Suggestion: Compare with at least three latest technologies
Reviewer 2 Report
The goal of this paper, as exposed by the authors, is to propose a novel Heterogeneous Parallel System Architecture with features of parallel computation and low power consumption.
The introduction is too short and does not present in detail the problems, current limitations and challenges of researchers regarding the requirements and scientific results. Its reading shows that it is a technical article of the use of the Industrial Ethernet technology and a comparison (it is located too early in the article). What Linux Embedded Hard Real-time EOS (Fig. 4) mechanisms uses the proposed Framework?
The hardware aspects related to Figure 9 and 12 must be present in the paper, validated with simulations and data. The oscilloscope captures (Figure 9 and 12) are not visible, and the following elements are not distinguished: Trigger, single/persistence mode, acquisition cycle.
Based on the information presented in this paper, what are the EtherCAT ISO/OSI model?
IEC 61158 protocol is suitable for hard real-time computing requirements in automation technology. The novel HPSA concept, proposed in this paper, is it also designed for firm real-time systems?
Section 6 is very concise and does not provide the expected innovation factor brought by the authors and future work directions.
The reference section is good, citing new and relevant articles in the research area.
Round 2
Reviewer 2 Report
The authors have addressed most of my concerns satisfactorily. Therefore, I propose the paper for publication.